# Challenges, Recent Advances and Perspectives in the Treatment of Human Cytomegalovirus Infections

**DOI:** 10.3390/tropicalmed7120439

**Published:** 2022-12-14

**Authors:** Shiu-Jau Chen, Shao-Cheng Wang, Yuan-Chuan Chen

**Affiliations:** 1Department of Neurosurgery, Mackay Memorial Hospital, Taipei 10449, Taiwan; 2Department of Medicine, Mackay Medical College, New Taipei City 25245, Taiwan; 3Department of Psychiatric, Taoyuan General Hospital, Ministry of Health and Welfare, Taoyuan 33004, Taiwan; 4Department of Mental Health, Johns Hopkins Bloomberg School of Public Health, Baltimore, MD 21205, USA; 5Department of Medical Technology, Jenteh Junior College of Medicine, Nursing and Management, Miaoli County 35664, Taiwan; 6Program in Comparative Biochemistry, University of California, Berkeley, CA 94720, USA

**Keywords:** human herpesvirus, cytomegalovirus, latency, acute/latent infection, gene targeting approach, cell therapy, compassionate use

## Abstract

Human cytomegalovirus (HCMV) is ubiquitous worldwide and elicits global health problems. The diseases associated with HCMV are a serious threat to humans, especially for the sick, infant, elderly and immunocompromised/immunodeficient individuals. Although traditional antiviral drugs (e.g., ganciclovir, valganciclovir, cidofovir, foscarnet) can be used to treat or prevent acute HCMV infections, their efficacy is limited because of toxicity, resistance issues, side effects and other problems. Fortunately, novel drugs (e.g., letermovir and maribavir) with less toxicity and drug/cross-resistance have been approved and put on the market in recent years. The nucleic acid-based gene-targeting approaches including the external guide sequences (EGSs)-RNase, the clustered regularly interspaced short palindromic repeats (CRISPRs)/CRISPRs-associated protein 9 (Cas9) system and transcription activator-like effector nucleases (TALENs) have been investigated to remove both lytic and latent CMV in vitro and/or in vivo. Cell therapy including the adoptive T cell therapy (ACT) and immunotherapy have been tried against drug-resistant and recurrent HCMV in patients receiving hematopoietic stem cell transplantation (HSCT) or solid organ transplant (SOT), and they have also been used to treat glioblastoma (GBM) associated with HCMV infections. These newly developed antiviral strategies are expected to yield fruitful results and make a significant contribution to the treatment of HCMV infections. Despite this progress, the nucleic acid-based gene-targeting approaches are still under study for basic research, and cell therapy is adopted in a small study population size or only successful in case reports. Additionally, no current drugs have been approved to be indicated for latent infections. Therefore, the next strategy is to develop antiviral strategies to elevate efficacy against acute and/or latent infections and overcome challenges such as toxicity, resistance issues, and side effects. In this review, we would explore the challenges, recent advances and perspectives in the treatment of HCMV infections. Furthermore, the suitable therapeutic strategies as well as the possibility for compassionate use would be evaluated.

## 1. Introduction

### 1.1. Structure and Classification of Human Herpesviruses

The human herpesvirus (HHV) is a spherical enveloped virus with a diameter ranged from 120 to 260 nm, usually about 150 nm. The genetic material of HHV contains a double-stranded DNA genome that encodes 100–200 genes in the decahedral capsid [1,2]. The herpesvirus replicates in its host cell nucleus, and its genes express in a highly coordinated transcriptional cascade as the following order: (1) immediate early genes, which code for regulatory proteins essential for viral proliferation and reactivation; (2) early genes, which code for enzymes essential for viral DNA replication; (3) late genes, which code for structural proteins essential for infectious viral particles (virions) assembly [1,2].

HHV can be divided into nine types: Herpes simplex virus-1 (HSV-1, HHV-1), Herpes simplex virus-2 (HSV-2, HHV-2), Varicella zoster virus (VZV, HHV-3), Nasopharyngeal virus (Epstein–Barr virus, EBV, HHV-4), Cytomegalovirus (HHV-5), Roseolovirus (HHV-6A, HHV-6B, HHV-7) and Kaposi’s sarcoma-associated herpesvirus (KSHV, HHV-8) [3,4].

### 1.2. Life Cycle of HHV

The life cycle of HHV can be divided into two types: lytic pathway and lysogenic pathway. In the lytic pathway, the virion attaches to a host cell with a specific receptor to initiate the process of infection. After the viral outer membrane-coated glycoprotein binds to the host’s cell membrane receptor, virions enter host cells by receptor-mediated endocytosis or membrane fusion [5]. After the viral capsid is decomposed, the genomic DNA is released. The invading DNA manipulates the host enzyme and energy system to synthesize new viral DNA and proteins and then assembles to form new virions. During the process of HCMV replication, the terminase complex cleaves DNA to package the genome into the capsid to complete DNA maturation and viral assembly [6,7,8]. Finally, the new virions release from original host cells to infect new hosts. However, in the lysogenic pathway, a few viral genes may transcribe latency-associated transcripts (LATs) in host cells. In this pattern, the virus can exist indefinitely without lysing host cells and cause no signs and symptoms in the host, resulting in long-term latency [3,9,10,11].

The latency is referred to the state in which the virus is dormant, and the latent infection is a persistent infection in which the viral DNA is present in the host cell. In this life cycle, viruses still can be transmitted to others in special cases (e.g., organ transplantation, congenital infection), although the virions cannot be detected and no obvious symptoms are found [4,9,10]. When the virus is stimulated or the host immune system is suppressed, the dormant viruses can be reactivated to produce large numbers of viral progeny to cause symptoms and diseases. There are two types of viral latency: proviral latency and episomal latency. In the provirus latency, the viral genomic DNA is incorporated in the host genome after the provirus enters the host cell, and the viral DNA replicate synchronously with the host DNA, such as human immunodeficiency virus (HIV). In the episomal latency, the viral DNA exists in the cytoplasm or nucleus of the host in a linear or lariat structure without integrating into the host genome. Thus, the virus still can utilize its own genetic materials, such as HHV [4,9,10]. After the latent HHV is reactivated, the transcription of the LAT gene would change to the lytic gene to enhance viral DNA replication and viral proliferation, which often causes non-specific initial symptoms clinically, such as fever, headache, sore throat, general malaise, rash, etc. These symptoms may worsen to cause severe diseases of patients, and in some severe cases, even lead to death [4,9,10].

### 1.3. Human Cytomegalovirus (HCMV) Molecular Biology and Infection

The HCMV belongs to genus Cytomegalovirus, subfamily Beta Herpesviridae, family Herpesviridae (HHV-5), which genome is a double-stranded DNA of about 230 kb in size and the virion diameter is about 180 nm. Like all HHV, the structure of HCMV includes a DNA core, capsid, an amorphous protein (tugument) and an outer membrane (envelope) from inside to outside. The DNA genome is surrounded by a capsid consisting of 162 capsomeres, and the tugument is surrounded by an envelope consisting of a lipid biolayer containing different glycoproteins (Figure 1) [11,12,13].

The HCMV genome encodes about 54 membrane proteins, and at least 25 membrane glycoproteins are found on the envelope [13]. HCMV infection requires a complex entry mechanism between viral glycoproteins and host receptors that depend on cell types which render a broad tropism to viruses. HCMV glycoproteins gB and gH/gL are required for viral entry [13,14,15,16]. The UL128, UL130 and UL131 proteins assemble onto gH/gL to produce pentameric complex gH/gL/UL128-131 and mediate the infection of epithelial and endothelial cells. The HCMV glycoprotein gO can facilitate the intracellular transport of gH/gL to the viral assembly site but unbinds prior to virions maturation. HCMV gO-mutants suggested a model in which gH/gL/UL128-131 mediates entry into epithelial/endothelial cells and a trimeric complex gH/gL/gO mediates entry into fibroblasts. If HCMV glycoproteins gO is absent, gH/gL in virions will be significantly decreased, and thereby, HCMV cannot enter fibroblasts, as well as epithelial and endothelial cells [13,14,15,16] (Figure 1).

## 2. Diseases Caused by HCMV Infections 

HCMV is mainly transmitted through blood, saliva, semen, urine, breast milk and vertical infection. The main target cells of infection are monocytes, lymphocytes and epithelial cells. After infection, HCMV has an ability to maintain latency within the body over long periods and may only result in mild or subclinical symptoms in immunocompetent adults [17,18]. The reactivation of latent HCMV destroys patient tissues to lead to organ diseases and may trigger immunomodulatory effects. Although CMV infection is usually asymptomatic in healthy people, it can cause opportunistic infections in individuals whose immune functions are compromised, immature or suppressed (e.g., older people, sick persons, immunosuppressed/immunodeficient patients, organ transplant recipients, etc.), leading to life-threatening consequences. Opportunistic infections resulting from latent infections can lead to a variety of diseases, including pneumonia, retinitis, gastrointestinal disease and cardiovascular disease, and even death in some severe cases [19,20,21]. For example, it is a common opportunist for transplant patients and a significant contributor to mortality for immunodeficient patients; it leads to increased hospitalization and pneumonia for intensive care patients, immunosenescence for the elderly, and increased mortality for the general population [22,23]. HCMV-associated diseases have become a main morbidity and mortality cause for humans [24,25]. In addition, other critical problems related to HCMV infection are described in the following subsection.

### 2.1. Congenital Infection for Infant

In a congenital infection, HCMV is transmitted directly from pregnant woman to the fetus through the placenta. Although most congenital infections are asymptomatic (85–90%), approximately 10–15% of neonates with congenital infections may have sequelae such as neurodevelopmental delays, mental retardation, jaundice, hepatosplenomegaly, small head deformities, hearing impairment, thrombocytopenia and other lifelong physical defects [26,27,28]. The most devastating one, the central nervous system sequelae, is related to neurodevelopment [29,30,31]. Because the damage to the central nervous system is irreversible and persistent, it may cause mental retardation, epilepsy, hearing loss, visual impairment, cognitive impairment, etc. Asymptomatic neonates showing congenital infection are at risk of long-term sequelae, especially mental retardation and sensorineural hearing loss for infant [29,30,31].

The HCMV congenital infection is one of the major causes of neonatal mental retardation and sensorineural hearing loss, which are lifelong defects that are impossible to detect through prenatal genetic testing. If the mother becomes infected with HCMV for the first time during pregnancy (primary infection), an intrauterine infection can occur and transmit HCMV to the fetus because she has already had on CMV antibodies. However, it is also possible that the mother already has CMV antibodies due to the reactivation of previous HCMV infections or has been infected with a different virus strain (non-primary infection). 

### 2.2. Glioblastoma (GBM) for Adults

Brain disease is a global human health problem that is often complex, severe, irreversible, persistent and refractory. The HCMV is a probable pathogen of brain diseases including glioblastoma (GBM). Certain malignant GBM expresses HCMV-related proteins, including the immediate early gene-1 product and the lower matrix phosphoprotein 65 (pp65) expressed by the unique long 83 (UL83) gene, indicating that HCMV may have played a role in the generation of GBM [32,33,34,35]. A certain animal study has revealed that CMV can induce GBM formation in vitro and in xenograft models and HCMV can also be detected in GBM isolated from patients, suggesting that HCMV may drive tumorigenesis or be reactivated in tumors [35]. However, there is no obvious and sufficient evidence to show that this phenomenon is pervasive in GBM formation currently [35].

The HCMV infection consequence for GBM is still controversial, because only very low levels of viral proteins and nucleic acids can be found in patients. However, the present studies have suggested a high HCMV infection rate in GBM patients [36,37,38,39]. Additionally, the mechanism of HCMV carcinogenesis has been explored in the oncomodulation, oncogenic features, tumor microenvironment regulation, epithelial–mesenchymal transition, overall immune system regulation, and other aspects [36,37,38,39].

## 3. Challenges for the Treatment of HCMV Infections

At present, antiviral drugs, nucleic acid-based gene-targeting approaches including external guide sequences (EGSs)-RNase, the clustered regularly interspaced short palindromic repeats (CRISPRs)/CRISPR-associated 9 (Cas9) nuclease system and transcription activator-like effectors nucleases (TALENs), and cell therapy (e.g., adoptive T cell therapy, immunotherapy) are the major strategies to treat HCMV, which have been applied clinically, under study for basic research, and used conditionally in limited patient population, respectively (Table 1) [4].

### 3.1. Traditional Antiviral Drugs Are Only Indicated for Acute Infections, Have Side Effects and Induce Resistance Issues

Some antiviral drugs including ganciclovir, valganciclovir, cidofovir, and foscarnet have been approved on the market and used clinically for the treatment of HCMV infections for years [4,40,41,42,43,44,45]. The mode of action of these traditional drugs is to inhibit the viral DNA polymerase [4,40,41,42,43,44,45] (Table 2). However, almost all traditional drugs must be injected intravenously and cannot be used in patients without side effects and causing other problems. That is, their efficacy is remains limited by toxicity, drug resistance, cross-resistance, and other issues; thus, the administration methods, effectiveness and safety need further improvement. Moreover, current antiviral drugs are only indicated for HCMV acute (active, productive, lytic) infections.

### 3.2. Gene-Targeting Approaches Have Difficulty in Delivery Tool Option and Safety Concerns 

The nucleic acid-based gene targeting approach includes strategies quite different from traditional antiviral drugs in the structure, mechanism, administration routes and delivery tools. They potentially provide effective strategies to treat diseases caused by HCMV infections by designing a specific DNA or RNA sequence to target essential genes for virus growth, such as EGSs-RNase [4,46,47,48], the CRISPRs/Cas9 nuclease system [49,50,51,52] and TALENs [4,53]. These unconventional antiviral approaches are probably promising in preventing HCMV infections or removing lytically and/or latently infected HCMV. Particularly, they may make a significant improvement to the eradication of HCMV latent infections. This should be the next highlight for developing unconventional antiviral approaches, because no existing drugs or therapeutic strategy have been approved for HCMV latent infections or both acute and latent infections. There have been many successful studies for inhibiting lytic replication and/or clear latent infections in previously published papers [46,47,48,49,50,51,52,53]. However, these approaches are all in vitro and/or in vivo studies, and none of them have been advanced to clinical trials. The critical challenges and major limitations are as follows: (1) Option of delivery tool: It is lyrically difficult to select an appropriate delivery tool that is safe, specific and efficient in delivering antiviral nucleic acids to target viruses or HCMV-infected cells in patients [54,55]. (2) Safety concern: The mutation and tumorigenesis may be caused by off target effects, which might target normal genes in patients [56,57,58].

### 3.3. Cell Therapy

#### 3.3.1. Adoptive T Cell Therapy for Transplantation Recipients

Adoptive T cell therapy (ACT), a kind of immunotherapy, involves the isolation and ex vivo expansion of specific T cells and expects to obtain more T cells than those induced by vaccination alone. The activation of T cell-mediated immune responses in transplantation recipients can decrease the morbidity and mortality caused by HCMV infections. The virus-specific T cells have potential efficacy to treat a variety of viruses including HCMV by targeting multiple pathogens in patients undergoing hematopoietic stem cell transplantation (HSCT) [59,60,61,62]. ACT has also been developed to combat a variety of drug-resistant and recurrent viruses including HCMV in solid organ transplant (SOT) recipients [59,60,61,62]. Hence, ACT is capable of being an antiviral strategy against CMV reactivation in patients who are receiving allogeneic HSCT and SOT [59,60,61,62]. However, the major limitations for ACT are as follows: (1) Efficacy evaluation is difficult: The HSCT and SOT recipients receiving ACT often also take other medicines because of complications or other diseases, and the concomitant treatments may confound the evaluating result of ACT. (2) Random controlled clinical trials are essential. Although the prospective, multicenter, open-label study has ensured the feasibility and safety of ACT, the clinical efficacy needs to be confirmed in random controlled trials with significant clinical statistics. (3) Prognosis manifestation is needed: The patients’ prognosis should be monitored for a long time to achieve definite evidence and results for clinical efficacy.

#### 3.3.2. Immunotherapy for GBM Patients

Because HCMV is often found to be present in GBM whereas it is absent from the normal brain, HCMV antigens are considered as appropriate immunological anti-GBM targets. The patient-derived HCMV pp65-specific T cells can recognize and kill autologous GBM tumor cells [37,63,64,65]. Hence, it is possible to develop immunotherapy to treat GBM, and some successful treatments have been reported in several cases [66,67,68]. However, the major limitations for GBM immunotherapy are as follows: (1) Efficacy evaluation is difficult: HCMV directed immunotherapy is affected by many factors such as heterogeneous HCMV expression in GBM, T cells immunosuppression in the tumor microenvironment, suppression mediated by inhibitory immune receptors on T cells, and the effect of prior standard treatment on the tumor and the host. To realize the potential of HCMV-directed immunotherapy completely, it is required to have a combination therapy and understand how to sequence this therapy [69]. (2) Prognosis manifestation is needed: The further follow-up studies must be performed and patients’ prognosis should be monitored over the long term to achieve definite evidence for clinical efficacy. (3) Clinical statistics must be significant: The clinical data support CMV antigen-directed immunotherapies against GBM, but almost all successful cases were only reported case by case. It is challenging to have a significant clinical statistical result.

## 4. Recent Advances and Perspectives

### 4.1. Novel Antiviral Drugs with Fewer Side Effects and/or Resistance Issues Have Been Developed or under Investigation

Two novel antiviral drugs have been approved for clinical application for the treatment of HCMV infections in recent years, such as letermovir and maribavir. Letermovir is used to prevent or treat acute HCMV infections and has been tested in infected patients and shown to be useful in organ transplant recipients or acquired immunodeficient syndrome (AIDS) patients, and thereby, it has been approved for the prophylaxis of HCMV in cell and/or organ transplantation recipients. Maribavir is approved for the treatment of adults with post-transplant HCMV infections who are refractory (with or without genotypic resistance) to one or more prior antiviral therapies. They can induce fewer side effects and less drug/cross-resistance, in addition to multiple administration routes (Table 2). 

Compared with traditional antiviral drugs (e.g., ganciclovir, valganciclovir, cidofovir, foscarnet), these novel drugs have the following advantages: (1) They can be taken orally, so patients do not need to be hospitalized and can administer drugs by themselves; (2) They are less toxic and have no critical side effects such as bone marrow toxicity and nephrotoxicity; (3) They acts on viral terminase complex or protein kinase rather than DNA polymerase, so there is no risk of cross-resistance with other antiviral drugs. However, their efficacy needs further verification, and data in prognosis and postmarket surveillance are not sufficient. More data are required for these new drugs to provide insights into viral mutations detected in patients, the interpretation of genotyping results, and the clinical relevance of results. Post-market surveillance systems and monitoring for HCMV data on drug/cross-resistance should also be well established. For example, the in vitro data have revealed that letermovir resistance mainly to the UL56 component of the terminase complex and uncommonly to UL89 has been triggered by viral mutations [70]. 

Moreover, several other drugs are being developed and preliminarily demonstrate efficient inhibition in the process of viral DNA replication and gene expression. Their anti-HCMV potential has been preclinically or clinically evaluated, such as brincidofovir, filociclovir and valnoctamide. However, these compounds need further studies and/or undergoing clinical trials before they can be approved for clinical application (Table 3). 

### 4.2. The Combination of Drugs Is Testing

Chou et al. show that maribavir can be used in combination with other HCMV antiviral drugs with an additive but not synergistic effects by the data of in vitro studies [89]. When maribavir was tested against wild-type and mutant viruses, it demonstrated additive interactions with foscarnet, cidofovir, letermovir and GW275175X as well as strong synergy with rapamycin, which suggest a potentially useful therapeutic combination [89]. However, it showed strong antagonism with ganciclovir, which is an effect that might resulted from the inhibition of pUL97-mediated ganciclovir phosphorylation [89].

Mercorelli et al. reported that a new extended-spectrum triazolic antifungal drug- isavuconazole (ICZ) had broad-spectrum activity against HCMV in vitro [90]. ICZ showed synergistic antiviral activity in vitro when administered in combination with other approved antiviral drugs including ganciclovir, foscarnet, and letermovir. The results suggested that future clinical studies should focus on evaluating the possibility of ICZ used in the combination treatment of HCMV infections [90]. 

The combination of antiviral drugs is a strategy to be used for the treatment of some HCMV infections, but their efficacy is still not clear. The efficacy of the combination of drugs is suggestive but not conclusive with very limited in vitro data compared to monotherapy [91]. More and further results of in vitro, in vivo and clinical trials are required to confirm that combination of drugs is superior to monotherapy, especially for the prevention or treatment of HCMV resistance. 

### 4.3. Gene-Targeting Approaches and Cell Therapy Are Feasible for Compassionate Use

Although nucleic acid-based gene targeting approaches have the potential to be directly used in human therapy, most of them are only at the stage of basic research or animal studies. None of them are actually being used in clinical trials or product manufacture because of the lack of specificity, concerns of mutations, and difficulty in delivery tool selection. For the treatment of HCMV infections, the gene-targeting approach may not be effective and even lead to aggravating the disease if traditional therapeutic methods are rashly ignored. However, there are exceptions, especially if the patients are at the urgent or dangerous phase of diseases and no effective drugs are available. In these seriously ill or life-threatening cases, the safety and ethics concerns of treatment based on gene-targeting approaches are less; thereby, compassionate use is possible. 

ACT is a strategy to restore immunological control against immunosuppressed patients affected by HCMV, and it seems to be promising for the prevention and therapy of HCMV infections in HSCT and SOT recipients. In addition, immunotherapy induces strong immune responses against GBM in patients with minimal damage to adjacent normal cells and healthy tissues. Overall, clinically successful cases provide further understanding about the role of HCMV in pathogenesis and support ACT for HSCT and SOT recipients and immunotherapy for GBM patients. However, these cases or trials are neither universal nor representative due to the small patient population size or case report status. Despite this, the compassionate use of cell therapy based on conditional approval is possible in case the diseases worsen and no existing effective drugs are available or the therapeutic benefits are considered to significantly outweigh possible risks in the preliminary evaluation.

Currently, it is a feasible pathway using unconventional antiviral strategies (e.g., nucleic acid-based gene targeting approaches and cell therapy) as compassionate use (expanded access) to treat HCMV infections such as letermovir, which was previously used in a case report [92] and a clinical trial program [93]. Compassionate use is probably advantageous for HCMV-infected patients with immediately life-threatening conditions or serious diseases to have access to investigational medical products including drugs, therapeutic approaches or medical devices whose clinical trials are still incomplete. The investigational medical products have not yet been approved to be safe and effective for their specific use and may cause unexpected side effects. However, they can be made available to patients who have no equivalent or satisfactory therapeutic options with the approval and supervision of an institutional review board (IRB) and the informed consent of patients. Compassionate use can be adapted if the requirements are consistent with the recommendations proposed by the United States of America Food and Drug Administration (USA FDA) and the European Medicines Agency (EMA) [94,95].

### 4.4. Further More Studies Have Been Conducted to Clearly Define the Efficacy and Risks of Cell Therapy 

For the past years, the adoptive transfer of HCMV-specific T cells has developed from the transfer of CD8 + T-cell clones to reduce the morbidity and mortality caused by HCMV infections in organ transplantation patients such as HSCT and SOT recipients [96] The incidence of HCMV reactivation may decrease, because a novel prophylactic antiviral drug, letermovir, is used as a prophylaxis in standard therapy. Therefore, any extensively adopted cell therapy needs to be more competitive and beneficial than standard therapy. The effective management strategies should be based on sensitive surveillance programs along with the introduction of letermovir; however, the object is to induce patient immune responses until adaptive immunity is sufficient to restrict viral replication [96]. The patients who are most likely to achieve the maximum advantages from cell therapy will be identified by analyzing data of clinical trials. Currently, the most successful treatment of HCMV infections using cell therapy is adopted in a small study population size or only a case report. Further and more studies and random controlled clinical trials are essential for the precise efficacy/risk evaluation, definite prognosis manifestation and significant statistical analysis [97,98]. 

### 4.5. HCMV Vaccine Is Being Developed

Farrell et al. explored the possibility of a mouse cytomegalovirus (MCMV) vaccine with the deletion of a viral G protein-coupled receptor (GPCR), designated M33, which is attenuated for systemic spread [99]. They used olfactory vaccination and virus challenge to resemble viral natural infection. A single non-invasive olfactory ΔM33 MCMV vaccine was replicated locally, but it was attenuated for latent infection and failed to spread systemically due to the loss of the M33 GPCR [99]. The results showed that a single olfactory MCMV cannot spread systemically and can protect mice from wild-type MCMV superinfection and transmission. This method of deleting functional genes in HCMV potentially provides a safe and effective vaccine against congenital infection [99].

Kschonsak et al. evaluated a key vaccine candidate, indicating that gH/gL/UL128-130-131A complex (pentamer) induces the most potent neutralizing response against HCMV [100]. The structural basis for the recognition of pentamer receptors and antibody neutralization is still poorly understood. They explored the structures of pentamer bound to neuropilin 2 and potent neutralizing antibodies against HCMV. In addition, they identify thrombomodulin (THBD) as a functional HCMV receptor and determine the structures of the pentamer–THBD complex [100]. The results revealed the framework of HCMV receptor engagement, the cell entry mechanism, and antibody neutralization, and they provide a strategy to against HCMV [100].

The objective of HCMV vaccine development should be the prophylaxis of congenital infection and systemic organ diseases in immunocompromised individuals (e.g., (AIDS and transplanted patients). However, the major barriers for the development of an effective HCMV vaccine are as follows: (1) The protection provided by immune memory cells against HCMV reactivation from latency and HCMV re-infection (infection of a seropositive individual by a new virus strain) is not sufficient [101,102]. (2) HCMV is able to escape from the immune response, unclear immune relationship for protection, insufficiency of available animal models, and lack of general awareness [103]. (3) It is difficult to determine which one is the best target population for HCHV vaccination [103].

## 5. Conclusions

There are still many people infected with HCMV in the world, although most of the infected people have no signs and symptoms. HCMV infects people of all ages, and over half of adults have been infected with HCMV by 40 years old [104,105,106,107,108,109]. It is called congenital infection when a baby is born with HCMV infections. About one out of 200 babies is born with congenital infection, and about one-fifth (1/5) of them will have long-term health problems [104,105,106,107,108,109]. By the suppression of the immune system, most people with HCMV infection are in latency and do not have obvious symptoms or diseases. However, serious symptoms and illnesses are likely to occur in highly risky groups such as infants, the elderly, sicks, organ transplanters, immunosuppressive agent users, immunodeficient/immunocompromised patients, etc. Therefore, we should pay attention to the HCMV-associated diseases and their impact on human health. It is essential to continue to explore more potential strategies and drugs for the treatment of HCMV acute and/or latent infections, although some antiviral drugs have been applied in treating HCMV infections clinically. Additionally, a feasible process for the effective treatment of HCMV infections should be established, including antiviral drugs, cell therapy, and nucleic acid-based gene targeting approaches (Figure 2).

## Figures and Tables

**Figure 1 tropicalmed-07-00439-f001:**
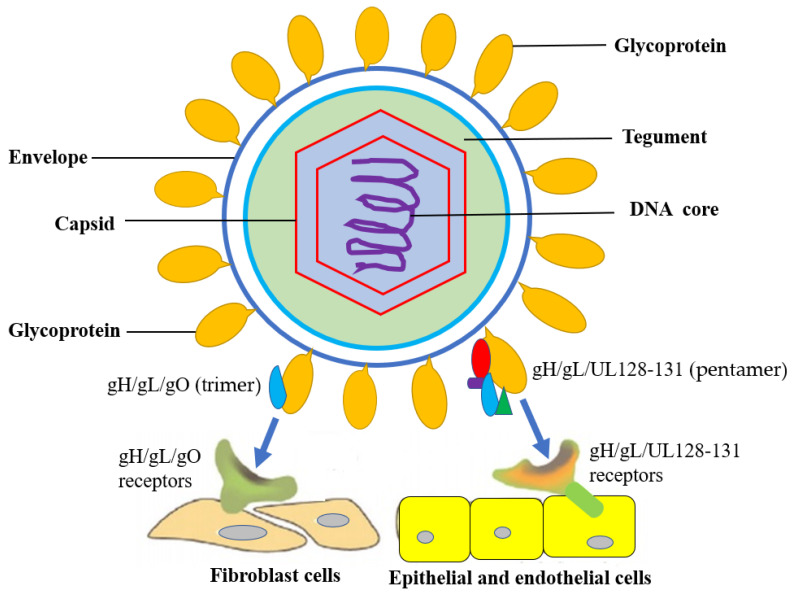
HCMV structure and entry into host cells. HCMV glycoproteins gH, gL, UL128, UL130 and UL131 assemble together to form a functional complex for binding to host cellular receptors, such as gH/gL/UL128- 131 receptors and gH/gL/gO receptors. gH/gL/UL128-131 mediates entry into epithelial/endothelial cells and gH/gL/gO mediates entry into fibroblasts.

**Figure 2 tropicalmed-07-00439-f002:**
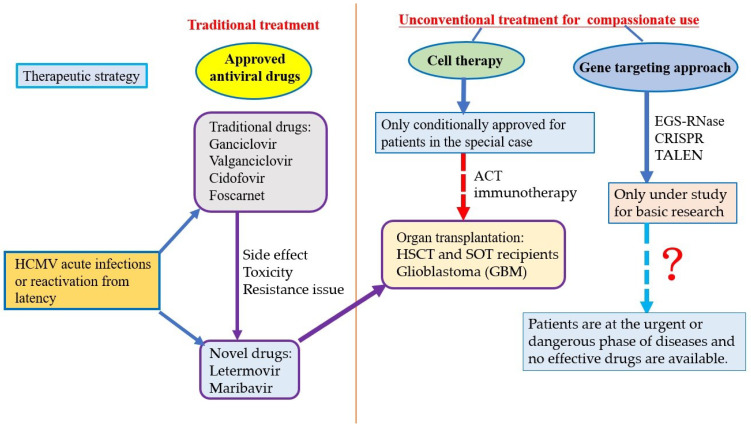
A feasible process using different strategies for the effective treatment of HCMV infections.

**Table 1 tropicalmed-07-00439-t001:** Major strategies have been used clinically or potential for the treatment of HCMV infections.

Strategy	Adoption	Current Status	Main Problem
Antiviral drugs	Traditional: ganciclovir, valganciclovir, cidofovir, foscarnet Novel: letermovir, maribavir	Extensively used clinically for the treatment of acute infections	Side effects, toxicity, resistance issues including drug resistance and cross resistance
Nucleic acid-based gene targeting approach	EGSs-RNase, CRISPRs/Cas9, TALENs	Only under study for basic research, not for clinical application so far	Option of delivery tool, safety concerns
Cell therapy	ACT, immunotherapy	Not universal, only used conditionally in limited patient population or special cases	Efficacy evaluation is difficult, Random controlled clinical trials and prognosis manifestation is still lacking

Abbreviation: External guide sequences (EGSs)-RNase; the clustered regularly interspaced short palindromic repeats (CRISPRs)/CRISPR-associated 9 (Cas9) nuclease system; transcription activator-like effectors nucleases (TALENs); adoptive T cell therapy (ACT).

**Table 2 tropicalmed-07-00439-t002:** Traditional antiviral drugs approved for the treatment of HCMV acute infections.

Drug	Structure	Mechanism	Main Side Effects
Ganciclovir	A synthetic analogue of 2′-deoxy-guanosine	Viral *UL54* DNA polymerase inhibitor	A potential carcinogen, granulocytopenia, neutropenia, anemia, thrombocytopenia
Valganciclovir	L-valyl ester of ganciclovir	Viral *UL54* DNA polymerase inhibitor	A potential carcinogen, granulocytopenia, neutropenia, anemia, thrombocytopenia
Cidofovir	A monophosphate nucleotide analogue	Viral *UL54* DNA polymerase inhibitor	Nephrotoxicity, neutropenia, nausea, uveitis, iritis, asthenia, alopecia, ocular hypotony
Foscarnet	Pyrophosphate analogue, a structural mimic of the anion pyrophosphate	Inhibitor of the pyrophosphate- binding site on viral DNA polymerase (or reverse transcriptase); Noncompetitive inhibitor of many RNA and *UL54* DNA polymerase	Nephrotoxicity, electrolyte disturbance, genital ulceration, paranesthesia, irritability, hallucination

**Table 3 tropicalmed-07-00439-t003:** Novel drugs have been approved or being developed for the treatment of HCMV infections.

Drug	Structure	Mechanism	Main Side Effects	Current Status	Reference
Letermovir	A non-nucleoside, 3,4-dihydroquinazolinyl acetic acid	Viral terminase complex inhibitor encoded by gene *UL56*, *UL51*, *UL89*	Nausea, diarrhea, vomiting, swelling in arms and legs, cough, headache, tiredness, hepatitis, stomach pain	Approved by the US Food and Drug Administration (FDA) and the European Medicines Agency (EMA) in 2017	[71,72,73,74,75,76]
Maribavir	A benzimidazole riboside	Viral protein kinase (*UL97*) inhibitor	Taste disturbance, nausea, diarrhea, vomiting and fatigue	Approved by USA FDA in 2021	[77,78,79,80]
Brincidofovir	An alkoxyalkyl ester prodrug containing the synthetic, acyclic nucleoside monophosphate analog cidofovir	Viral *UL54* DNA polymerase inhibitor	Diarrhea, nausea, vomiting, and abdominal pain and others (under study)	Under clinical trial	[76,81,82,83]
Filociclovir	A guanosine nucleoside analog	Viral *UL54* DNA polymerase and the *UL97* kinase inhibitor	Mild to extreme stomach upset, headaches, mild fever and others (under study)	Under clinical trial	[84,85,86]
Valnoctamide	A structural isomer of valpromide, a valproic acid prodrug	Inhibition of viral attachment to the host cell	Sommolence, the slight motor impairments and others (under study)	Under study	[87,88]

## Data Availability

This is a review article and does not report any data.

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
