# Peer review of "Challenges, Recent Advances and Perspectives in the Treatment of Human Cytomegalovirus Infections"

_tropicalmed, 2022, doi:10.3390/tropicalmed7120439_

Round 1
Reviewer 1 Report
This review has updated information about the latest treatments for HCMV infections, and therapeutic strategies. I recommend checking the grammar and the style, look at the bibliography as there is some word in different color.
Author Response
Comments and Suggestions for Authors:
This review has updated information about the latest treatments for HCMV infections, and therapeutic strategies. I recommend checking the grammar and the style, look at the bibliography as there is some word in different color.
Ans: We have checked the grammar, the style and the bibliography to revise them all over the manuscript.
Reviewer 2 Report
Chen SJ et al in article titled `Challenges, Recent Advances and Perspectives and Treatment of human Cytomegalovirus Infection` describe consequences of the CMV to population at risk, standard and novel treatment approaches.
The article is well written and organized. The tables and the Figure are of a good quality.
The good side of the article is that gives exact description of the need for the new treatment for the population at risk treatment (standard with its flaws versus novel already approved by FDA and EMA), but also a need to speed up an improvement of future drugs and treatments.
The thing that could be added to have stronger impact is a bit more of the statistical data of population-at-risk infected with CMV.
Minor grammatical and typo lines : 184, 185, 255, 276, 304, 305
Author Response
Comments and Suggestions for Authors:
Chen SJ et al in article titled `Challenges, Recent Advances and Perspectives and Treatment of human Cytomegalovirus Infection` describe consequences of the CMV to population at risk, standard and novel treatment approaches. The article is well written and organized. The tables and the Figure are of a good quality. The good side of the article is that gives exact description of the need for the new treatment for the population at risk treatment (standard with its flaws versus novel already approved by FDA and EMA), but also a need to speed up an improvement of future drugs and treatments. The thing that could be added to have stronger impact is a bit more of the statistical data of population-at-risk infected with CMV.
Minor grammatical and typo lines : 184, 185, 255, 276, 304, 305
Ans: We have added a paragraph to describe the statistical data of population-at-risk infected with CMV. ( line 394~399 in P.12) (highlighted in yellow). Additionally, we have revised the minor grammatical errors and typos on line 184, 185, 255, 276, 304, 305.
Reviewer 3 Report
HCMV can cause serious diseases in immunocompromised patients. Current antiviral inhibitors (ganciclovir, cidofovir and foscarnet) all target the viral DNA polymerase. They have adverse effects and prolonged treatment can select for drug resistance mutations. Thus, we need new drugs.
The manuscript is well written. The review is highly relevant to the field.
Comments:
1. Since Letermovir target the terminase complex, authors should extend the DNA packaging step of the viral life cycle. What about letermovir mechanism of action? The authors should cite Heming et al., 2017 and Ligat et al., 2018. And cite also resistance mutation observed on terminase subunits.
2. What about combination of drugs?
3. The authors should add a section on vaccine.
Author Response
Comments and Suggestions for Authors:
HCMV can cause serious diseases in immunocompromised patients. Current antiviral inhibitors (ganciclovir, cidofovir and foscarnet) all target the viral DNA polymerase. They have adverse effects and prolonged treatment can select for drug resistance mutations. Thus, we need new drugs. The manuscript is well written. The review is highly relevant to the field.
Comments:
- Since Letermovir target the terminase complex, authors should extend the DNA packaging step of the viral life cycle. What about letermovir mechanism of action? The authors should cite Heming et al., 2017 and Ligat et al., 2018. And cite also resistance mutation observed on terminase subunits.
Ans: The mechanism of action of letermovir is to inhibit the the terminase complex. Thus, we have added a sentence to describe the function of the terminase complex during HCMV replication including references. (line 63-65, P.5) (highlighted in yellow). Also, we have added a sentence to describe the resistance mutation observed on terminase subunits including reference (line 278-280, P.8) (highlighted in yellow).
- What about combination of drugs?
Ans: We have added a subsection 4.3 to describe the combination of drugs including reference (line 288-307, P.9-10) (highlighted in yellow).
- The authors should add a section on vaccine.
Ans: We have added a subsection 4.5 to describe the HCMV vaccine including references (line 394-399, P.12) (highlighted in yellow).